# Impact of Heating Rates on *Alicyclobacillus acidoterrestris* Heat Resistance under Non-Isothermal Treatments and Use of Mathematical Modelling to Optimize Orange Juice Processing

**DOI:** 10.3390/foods10071496

**Published:** 2021-06-28

**Authors:** Juan-Pablo Huertas, María Ros-Chumillas, Alberto Garre, Pablo S. Fernández, Arantxa Aznar, Asunción Iguaz, Arturo Esnoz, Alfredo Palop

**Affiliations:** 1Departamento de Ingeniería Agronómica, Instituto de Biotecnología Vegetal, Universidad Politécnica de Cartagena, 30203 Cartagena, Spain; jphuertasb@gmail.com (J.-P.H.); may.ros@upct.es (M.R.-C.); alberto.garreperez@wur.nl (A.G.); arantxa.aznar@upct.es (A.A.); asun.iguaz@upct.es (A.I.); arturo.esnoz@upct.es (A.E.); alfredo.palop@upct.es (A.P.); 2Food Microbiology Group, Wageningen University & Research, P.O. Box 17, 6700 AA Wageningen, The Netherlands

**Keywords:** predictive microbiology, food safety, thermal resistance, dynamic modeling, pasteurization

## Abstract

*Alicyclobacillus acidoterrestris* is a spoilage microorganism responsible for relevant product and economic losses in the beverage and juice industry. Spores of this microorganism can survive industrial heat treatments and cause spoilage during posterior storage. Therefore, an effective design of processing treatments requires an accurate understanding of the heat resistance of this microorganism. Considering that industrial treatments are dynamic; this understanding must include how the heat resistance of the microorganism is affected by the heating rate during the heating and cooling phases. The main objective of this study was to establish the effect of heating rates and complex thermal treatments on the inactivation kinetics of *A. acidoterrestris.* Isothermal experiments between 90 and 105 °C were carried out in a Mastia thermoresistometer, as well as four different dynamic treatments. Although most of the inactivation takes place during the holding phase, our results indicate the relevance of the heating phase for the effectiveness of the treatment. The thermal resistance of *A. acidoterrestris* is affected by the heating rate during the heating phase. Specifically, higher heating rates resulted in an increased microbial inactivation with respect to the one predicted based on isothermal experiments. These results provide novel information regarding the heat response of this microorganism, which can be valuable for the design of effective heat treatments to improve product safety and stability. Moreover, it highlights the need to incorporate experimental data based on dynamic treatments in process design, as heating rates can have a very significant effect on the thermal resistance of microorganisms.

## 1. Introduction

The intrinsic characteristics of juice and beverage products (pH < 4.1, with high content of organic acids and sugars) are a challenge for the survival and growth of most bacteria, so the most relevant foodborne microorganisms for the shelf life of this type of products are mostly acidolactic bacteria, and some yeasts and molds [1,2,3]. Juice and beverage commercial products are generally thermally treated by pasteurization processes (typically 88–90 °C for 2 min or 90–95 °C for 30–60 s), which inactivate the microbial flora and enzymes of these products. The outgrowth of the microbial cells surviving the treatment (mostly bacterial spores) is inhibited by the product characteristics [4,5], ensuring the stability of the product in most cases. Nevertheless, some spore-forming bacteria may still be able to grow during storage despite the low pH.

One of the main concerns for pasteurized juice products is *Alicyclobacillus acidoterrestris.* It is a Gram-positive, thermoacidophilic, spore-forming, spoilage bacterium. It grows at temperatures ranging from 26 to 60 °C (optimum between 42 and 53 °C) and pH values from 2.0 to 6.0 (optimum between 3.5 and 5.0) [5,6,7]. *A. acidoterrestris* spores can survive pasteurization treatments applied to juice and beverage products, and they may germinate due to the heat shock applied during the treatment. The surviving vegetative cells of *A. acidoterrestris* can grow at low pH values, making it one of the main concerns for the shelf life of this kind of product [2,8,9].

A wide range of high-acid, shelf-stable, hot-filled, pasteurized, canned, ultra-heat treated, or carbonated products have been spoiled by *Alicyclobacillus* spp. [5,9]. This spoilage, even with small numbers of viable microorganisms, could contaminate large volumes of products, inflicting significant and severe economic losses for juice processors [6]. Considering the product intrinsic characteristics, as well as the typical process conditions, *A. acidoterrestris* spores have been proposed as target microorganisms for juice and beverage products for the design or development of preservation processes [6,10,11].

In spite of the development of novel technologies during the last years, heat processing is still the most common method for pasteurization [12]. Although this technology is effective at inactivating spoilage and pathogenic microorganisms, it can also have a negative impact on the quality of most food products, colliding with consumer demands for minimally processed products. An optimum compromise between quality and safety/stability can only be achieved through a minimum treatment, which fulfills the desired microbial inactivation with a minimum impact on product quality. The design of such treatment without incurring microbial losses requires detailed knowledge of the microbial response during the thermal treatment, usually reflected in mathematical models [13]. This is the task of predictive microbiology, which uses mathematical models to predict the microbial response (outgrowth or inactivation) during the different stages of the life cycle of the food product. A mathematical model represents a real system using differential and/or algebraic equations that represent its more significant properties [14]. The application of mathematical models to describe microbial inactivation allows us to generate more accurate risk assessment procedures and quality assurance [15].

Predictive microbiology has been extensively used for the design of effective thermal treatments during the last decades. Nevertheless, in most cases, this has been done based on models fitted to experimental data gathered under isothermal conditions. However, because pasteurization temperatures are much higher than room temperature, industrial treatments involve (at least) three distinct stages: heating, holding, and cooling. The impact of these three stages should ideally be included in the process design mainly for two reasons. The first one is that every stage contributes to the total microbial inactivation, even if the temperatures during the heating and cooling phases are lower than the ones during the holding phase. More importantly, plenty of scientific evidence points out that the heating phase can have a strong influence on the thermal resistance of the microbial cells during the holding phase [16,17,18,19,20]. Because of this, the predictions of the model based on isothermal data may be biased for dynamic conditions, as these experimental settings cannot represent the effect of different heating rates.

There is a vast number of published results in a wide variety of heating media and temperatures on the behavior and thermal resistance of *A. acidoterrestris* under isothermal treatment conditions [5,9,10,21,22], but little is known when non-isothermal conditions are applied. There are only a handful of works about the effect of heating rates on the microbial spore inactivation kinetics and microbial populations on complex thermal treatments. Two previous works within our group studied the heat resistance of *A. acidoterrestris* under dynamic conditions [23,24]. However, these studies were mostly descriptive and did not perform a quantitative analysis of the effect of the different temperature profiles on the heat resistance based on mathematical models. Therefore, the objective of this study was to assess whether the heat resistance of *A. acidoterrestris* can be predicted based on models gathered under isothermal conditions, or if the shape of the thermal profile (e.g., the heating rate) affects the thermal resistance of this microorganism in order to optimize orange juice processing.

## 2. Materials and Methods

### 2.1. Microorganism

*A. acidoterrestris* DSM 3922 was provided by the German collection of microorganism and cell cultures (DSMZ). Sporulation was carried out in Petri dishes of Potato Dextrose Agar (PDA, Scharlau Chemie, Barcelona, Spain) by inoculating the agar surface with 0.2 mL of a 24 h culture grown at 42 °C in *Alicyclobacillus* spp. medium (BAT; Döhler, Darmstadt, Germany). The concentration of the spores was adjusted to 10^9^ spores mL^−1^ with sterile bi-distilled water. The spore suspension was stored at 0–5 °C until used.

### 2.2. Heat Resistance Determinations

Every heat resistance determination was performed in a thermoresistometer Mastia [24], which can reproduce isothermal or non-isothermal treatments with different heating profiles and rates. The temperatures of the isothermal heat treatments were 90, 95, 100, and 105 °C. For non-isothermal treatments, the thermoresistometer was programmed to perform the following monophasic profiles: at 1 °C/min with an initial temperature of 80 °C and a final one of 100 °C; and at 20 °C/min with an initial temperature of 80 °C and a final one of 110 °C. Furthermore, more complex heat treatments with three stages (heating, holding, and cooling) and two stages (heating and cooling) were also performed. For the three-stage treatment, the thermoresistometer was programmed to start at an initial temperature of 70 °C, which was increased to 105 °C at a heating rate of 27 °C/min. This temperature was held for 60 s, and finally, it was cooled down to 70 °C, at a cooling rate of 15 °C/min. For the two-stage treatment, the thermoresistometer was programmed to start at an initial temperature of 80 °C and reach a final temperature of 110 °C with a heating rate of 30 °C/min. Immediately after the final treatment temperature was reached, it was cooled down to 80 °C at a cooling rate of 30 °C/min.

For every heat treatment, the vessel of the thermoresistometer was filled with 400 mL of the heating medium. The heating media were pasteurized commercial orange juice, pH 3.5 and 11.0° Brix (García Carrión, Jumilla, España), pH 3.5 phosphate citrate McIlvaine buffer [25] and acidified Peptone Water (PW; 10 g/L peptone from casein (Scharlau, Chemie) and 5 g/L NaCl (Panreac, Barcelona, Spain)) with a final pH of 3.5 ± 0.1. Peptone water was acidified with Citric Acid (1 N) (Panreac, Barcelona, Spain). Acidified PW was sterilized prior to bacterial inoculation in the thermoresistometer and then adjusted to the treatment temperature. When orange juice was used as the heating medium, the instrument was sterilized with distilled water, cooled, emptied, immediately filled with commercially sterile orange juice, under sterile conditions, and heated to the treatment temperature. Once the target temperature was reached, the medium was inoculated with 0.2 mL of the spore suspension, resulting in an initial concentration of ~5 × 10^5^ CFU mL^−1^.

Samples were collected into sterile test tubes at preset time intervals, which were cooled on ice water immediately. All samples were then appropriately diluted, plated and plates were incubated at 42 °C for 24 h. Three separate experiments per condition were performed. Viable plate counts were based on duplicate counts, from appropriate dilutions plated in PDA.

### 2.3. Mathematical Models of Microbial Inactivation

The microbial inactivation of *A. acidoterrestris* was described using the Bigelow and Mafart models. The Bigelow model [26] can be considered the predecessor for inactivation models in predictive microbiology. It assumes that the microbial inactivation for a constant heating temperature follows a first order kinetics reaction. Therefore, a linear relation exists between the logarithm of the number of microorganisms (N) and the treatment time (t) as shown in Equation (1).
(1)$$\log_{10}N\left(t\right)=\log_{10}N_{0}-\frac{1}{D_{T}}t$$

The parameter $D_{T}$, usually called D-value, represents the inactivation rate being equal to the treatment time required to inactivate 90% of the microbial population at any value of *t*. $N_{0}$ stands for the initial number of microorganisms. It is considered that the D-value follows a log-linear relationship with temperature, as shown in Equation (2), where $D_{R}$ is the *D*-value calculated at the reference temperature ($T_{R}$). The reference temperature has no biological interpretation, although it can affect parameter identifiability [27]. Its value was set to 95 °C for every condition. The z-value (*z*) quantifies the sensitivity of the microorganism to temperature changes, being equal to the temperature increase required to cause a ten-fold reduction of the D-value.
(2)$$\log_{10}D_{T}=\log_{10}D_{R}+\frac{T_{R}-T}{z}$$

Equation (1) can be adapted for non-isothermal conditions by considering that the instantaneous rate of decay will be identical to the isothermal one. This hypothesis can be expressed as an ordinary differential equation as shown in Equation (3). The variation of the D-value with temperature is also expressed by Equation (2) for the dynamic case.
(3)$$\frac{d\left(\log_{10}N\left(t\right)\right)}{dt}=\frac{-1}{D_{T}}$$

The Mafart model [28] is based on the hypothesis that the time that each member of a microbial population can resist a heat stress follows a Weibull distribution. Under isothermal conditions, this hypothesis can be represented as shown in Equation (4), where *δ_T_* is the *δ*-value at temperature *T*, which represents the treatment time required to cause the first log-reduction of the microbial population. The curvature of the Mafart model is defined by parameter *p*. When *p* < 1, the survivor curves have upward curvatures, whereas they have a downwards curvature when *p* > 1. In the particular case where *p* = 1, the survivor curves are linear and the Mafart and Bigelow model are equivalent.
(4)$$\log_{10}N\left(t\right)=\log_{10}N_{0}-{\left(\frac{t}{\delta_{T}}\right)}^{p}$$

The parameter *p* is usually considered temperature independent, whereas a log-linear relationship between $\delta_{T}$ and temperature is considered (Equation (5)). The sensitivity of $\delta_{T}$ to temperature variations is quantified by the z-value (*z*), in the same way as in the Bigelow model.
(5)$$\log_{10}\delta_{T}=\log_{10}\delta_{R}+\frac{T_{R}-T}{z}$$

The Mafart model can be adapted for dynamic conditions under the hypothesis that the instantaneous inactivation rate is equal to the one observed during isothermal conditions [29]. This results in the differential equation shown in Equation (6), where *p* is constant and the relationship between $\delta_{T}$ and temperature is given by Equation (5).
(6)$$\frac{d\left(\log_{10}N\left(t\right)\right)}{dt}=-p\cdot \frac{t^{p-1}}{\delta_{T}^{p}}$$

### 2.4. Data Analysis

Model fitting and predictions under dynamic conditions were done using the functions included in the *bioinactivation* R package [30,31]. Both the Bigelow and Mafart models were fitted to the data gathered under isothermal conditions using a one-step approach based on non-linear regression with the Newton–Raphson algorithm. Initial guesses for the parameter estimates were defined based on preliminary simulations, and several values were tested without observing any relevant impact on parameter estimates. The dynamic data was fitted using an Adaptive Monte Carlo algorithm [32]. The convergence of the algorithm was evaluated using common guidelines using trace or pair plots [33], requiring 4000 iterations without burning length for convergence.

The parameters of the models estimated based on isothermal or dynamic experiments were used to predict the microbial response under dynamic conditions. These values were substituted in Equations (3) and (6), which were solved numerically using the LSODA algorithm [34], using the functions implemented in *bioinactivation*. These predictions were based on the temperature recordings in the media measured by the Mastia thermoresistometer.

The evaluation of the model fit and model selection were based on statistical indexes commonly used in predictive microbiology: the Root Mean Squared Error (*RMSE*) and the Akaike Information Criterion (*AIC_c_*). The RMSE quantifies the overall difference between the model fit/predictions and the observations, with values closer to one indicating a better correspondence. It is described in Equation (7), where *n* stands for the number of observations, $N_{i}$ are the microbial concentration observed, and ${\hat{N}}_{i}$ are the model predictions/fits. The *AIC_c_* (Equation (8)) is commonly used for model selection, as it also accounts for model parsimony (the number of model parameters, *p*). Models with a lower *AIC_c_* are preferred over models with higher *AIC_c_*.
(7)$$RMSE=\sqrt{\frac{1}{n}\sum {\left(log_{10}N_{i}-log_{10}{\hat{N}}_{i}\right)}^{2}}\;\;\;\;\;$$
(8)$$AIC_{c}=2k+n\cdot ln\left(\frac{\sqrt{\frac{1}{n-p}\sum {\left(log_{10}N_{i}-log_{10}{\hat{N}}_{i}\right)}^{2}}}{n}\right)-2\;\cdot p\frac{k+1}{n-k-1}$$

The accuracy and bias of the model predictions with respect to the data gathered under dynamic conditions were based on the accuracy *A_f_* and bias *B_f_* factors [35]. The accuracy factor (Equation (9)) is similar to the RMSE, quantifying the spread of the error term, with an accuracy factor of one indicating perfect agreement between the model prediction and the data. The bias factor (Equation (10)) indicates if there is a systematic deviation of the model predictions with respect to the experimental data. A bias factor equal to one, indicates a perfect agreement, whereas higher values correspond to an underestimation of the number of the microbial count and lower values an overestimation.
(9)$$A_{f}=e^{{\left(\sum \frac{1}{n}{\left(ln{\hat{N}}_{i}-lnN_{i}\right)}^{2}\right)}^{1/2}}$$
(10)$$B_{f}=e^{\frac{1}{n}\sum \left(ln{\hat{N}}_{i}-lnN_{i}\right)}$$

## 3. Results and Discussion

### 3.1. Thermal Resistance of Alicyclobacillus Acidoterrestris under Isothermal Treatment Conditions

Figure 1 shows the survival curves of *A. acidoterrestris* spores under the different isothermal treatments tested in acidified peptone water (pH 3.5). The survival curves have only a slight curvature at every temperature evaluated. This is reflected in the correct fitting of the Bigelow model, and the value of the shape parameter (*p*) of the Mafart model close to one. For the Bigelow model, the estimated parameters for *A. acidoterrestris* based on a one-step fitting were D_95_ = 5.24 ± 0.17 min and z = 11.2 ± 0.27 °C, whereas for the Mafart model, the estimated parameters were *p* = 1.20 ± 0.09, δ_95_ = 5.72 ± 0.26 min, and z = 11.4 ± 0.22 °C. For this situation, where the shape parameter of the Mafart model is close to one, the δ-value and the D-value have a similar interpretation.

Indeed, no statistical differences exist (*p* < 0.05) between the estimated values of D_95_ and δ_95_, nor between the z-values estimated using either model, highlighting the similarity for the predictions under isothermal conditions.

The effect of the heating medium on the thermal resistance of *A. acidoterrestris* under isothermal conditions has been thoroughly studied [2,5,36]. Tianli et al. [5] observed that treatment temperature, pH, soluble solids content (SSC), strain, medium, and divalent cations are the main factors that affect the thermal resistance of *A. acidoterrestris* spores. For different combinations of these factors, the reviews of Merle and Montville (2014) and Tianli et al. [5] reported D_95_ values between 0.06 and 5.3 min and between 1 to 10 min, respectively. With respect to the z-value, observations ranging between 6 and 22 °C have been observed for juice and beverages, whereas for laboratory media (buffers), they range between 5 and 10 °C [5]. Following a meta-analysis of published data on the effect of temperature and pH on the heat resistance of *A. acidoterrestris*, Silva et al. [21] calculated a D_95_ value of 4.9 min and a z value of 12.48 °C in pH 3.5 single strength orange juice, which falls very close to our findings. Conesa et al. [24] and López et al. [23] have determined the thermal resistance of this same strain of *A. acidoterrestris* in orange and tangerine juice, respectively. The reported D_95_ values of both studies were slightly higher than the estimated D_95_ in this research, whereas the z-values obtained in this research are slightly higher than the ones reported by these researchers. Previous research has shown that the D- and z-values obtained in acidified peptone water are usually higher than those reported on other laboratory media, but closer to those reported for juice and beverages. The thermal resistance of this microorganism in other laboratory media, such as pH 3.5 McIlvaine buffer, showed to be lower and this could be due to the absence of protective compounds (i.e., proteins, fats, fiber) since it is only composed of salts [5,23,36,37]. Peptone water is a laboratory medium that has some components that could act as protective agents.

### 3.2. Prediction of Non-Isothermal Inactivation Based on Isothermal Data

In order to evaluate whether the dynamic temperature profiles affect the thermal resistance of *A. acidoterrestris*, the predictions of the models fitted to isothermal data were compared against the observations gathered under dynamic conditions. Figure 2A,B compare the model predictions based on isothermal data against the observed inactivation of *A. acidoterrestris* spores under non-isothermal treatments at constant heating rates (1 and 20 °C/min) in acidified peptone water (pH 3.5). Figure 2C,D make the same comparison for the complex thermal treatments in acidified peptone water (pH 3.5) and orange juice. Regarding the effect of the media, no significant differences (*p* < 0.05) were observed between the results obtained in orange juice and acidified peptone water (pH 3.5), confirming the adequacy of this medium for the simulation of microbial inactivation in this food matrix. The plots show that the predictions based on isothermal data (using either the Bigelow or Mafart models) are biased in most cases with respect to the experimental data for every condition tested. Similar deviations between predictions based on isothermal data and dynamic observations have been reported in the literature [18,20,38], and indicate that the dynamic thermal profile affects the thermal resistance of the microorganism in a way that cannot be observed in isothermal experiments. The bias of the model predictions based on isothermal data is quantified in Table 1. There is a relatively good agreement between the model predictions and the observations for the temperature profile with the slowest heating rate (1 °C/min), illustrated by a bias factor close to one (Bf = 1.33 for the Bigelow model, Bf = 0.74 for the Mafart model). This agrees with the results reported by Conesa et al. [24], who also observed a good agreement between predictions based on isothermal data and dynamic experiments for a heating rate of 1 °C/min. At this slow rate, spore activation may take place, as treatments of 80 °C for 10–20 min are typically used prior to germination studies although this will not affect heat resistance [39]. Germination takes place in activated spores that are exposed to germinants or growth media in optimal conditions [40,41], which were not met in this study.

On the other hand, a high, positive B_f_ (the model overpredicts the microbial count) is observed for the other profiles tested, all of which have a heating rate higher than 20 °C/min. Based on this information, we can conclude that there is an effect on the heating rate on the thermal resistance of *A. acidoterrestris*. Namely, the application of heating rates between 20 and 30 °C/min reduced the thermal resistance of the microorganism with respect to the one observed under isothermal conditions. Nonetheless, this effect faces lower heating rates (1 °C/min). The effect of fast heating rates on the reduction of thermal resistance could also be somehow perceived in the former research by Conesa et al. [24]. Esteban et al. [42] also found accurate predictions for *B. sporothermodurans* spores exposed to slow heating rates (1 °C/min) in different heating media but failed to predict at faster heating rates (10 °C/min). Similar results were obtained with different spore-forming microorganisms by Gómez-Jódar et al. [43]. These results agree with the ones reported here.

It is worth highlighting the relative importance that each phase of the thermal treatment has on the total microbial inactivation for the complete treatment. In this case, the heating phase takes 28.1% of the total treatment time, the holding phase a 21.6%, and the cooling stage a 50.3%. However, the relative weight of each phase to the total inactivation is not proportional to their duration. Due to the exponential relationship between the D (or delta)-value and temperature, the inactivation achieved during the heating and cooling phases are negligible with respect to the one reached during the holding phase. These results point that, even when the holding stage has the smallest proportion of the treatment time, this stage represents the critical control point for microbial inactivation. Therefore, it is paramount for the design of effective heat treatments to accurately describe the microbial response during this phase. Nevertheless, studies should not be solely focused on describing the isothermal inactivation during the holding phase. As shown here (as well as in previous studies), the heating phase can have a strong influence on the microbial resistance during the holding phase, increasing or reducing it depending on several factors (e.g., microorganism, media, heating rate). This raises the need for further research studying how the heat resistance of foodborne pathogens is affected by dynamic treatments, such as those applied in industry.

### 3.3. Estimation of the Thermal Resistance and the Effect of the Heating Rates from Non-Isothermal Data

The results of the previous section have shown that, for some conditions, inactivation models are unable to predict the microbial response during dynamic treatments. Nevertheless, the results of the previous section (e.g., Figure 2, Table 1) are mostly qualitative and do not provide a quantitative estimate of the effect of the heating rate on the thermal resistance of the microorganism. For that reason, in this section, we fit the inactivation models (Bigelow and Mafart) directly to the data gathered under dynamic conditions using *bioinactivation*. Table 2 and Table 3 report the estimated parameter values for the Bigelow and Mafart models, respectively. Both models were able to fit the dynamic data, as illustrated in Figure 2, and as shown by the relatively low RMSE for every case (<0.7 in every case).

The parameter estimates in Table 2 reinforce the conclusions of the previous section. The monophasic profile with a heating rate of 1 °C/min resulted in a $D_{95}$-value of 5.77 min and a z-value of 11.47 °C. These values are similar to the ones estimated based on isothermal experiments (5.24 min, 11.4 °C), showing that the thermal resistance of the microorganism is similar in both conditions. However, the remaining conditions had a reduction in the D_95_ with respect to the one observed under isothermal conditions (22% reduction for the monophasic profile at 20 °C/min, 54% reduction for the heating + cooling profile, and 35% reduction for the complex profile). The z-value, on the other hand, showed little variation with respect to the one estimated under isothermal conditions. Similar results were observed for the Mafart model (Table 3). The slower heating rate (1 °C/min) resulted in a $\delta_{95}$-value of 5.06 min, close to the one estimated under isothermal conditions (5.72 min).

Higher heating rates resulted in larger variations of the $\delta_{95}$-value, with a reduction of the 40% for the monophasic profile at 20 °C/min, of the 60% for the heating + cooling one, and of the 32% for the complex profile. Nevertheless, as well as for the Bigelow model, no significant variation was observed in the estimated z- or *p*-values.

Table 2 and Table 3 report the values of *AIC_c_* calculated for each curve fitted to the experimental data. In every case, the *AIC_c_* calculated for the Bigelow model is lower than the one calculated for the Mafart model. This implies that the inclusion of one extra model parameter in the Mafart model is not justified when the model accuracy is compared to that of the Bigelow model for the non-isothermal inactivation of *A. acidoterrestris*. This may seem counterintuitive, due to the high non-linearity of the non-isothermal inactivation curves. However, this shape is caused by the existing log-linear relationship between the inactivation rate and the treatment temperature, which is considered by the Bigelow model. Therefore, in this case, the dynamic data can be described by the simpler Bigelow model.

The models fitted to the dynamic data serve to confirm the conclusions drawn in the previous section based on models gathered under isothermal data: high heating rates reduce the thermal resistance of the microorganism. Moreover, they provide additional insight on how this change in the thermal resistance takes place. According to Table 2 and Table 3, the different types of profiles affect the value of D_95_, whereas the z-value remains practically unchanged. Therefore, the secondary model is “shifted” in the *y*-axis. Because of this, the models were fitted again to the dynamic data fixing the z-value; i.e., estimating only the D_95_ (Bigelow model) or $\delta_{95}$ and *p*-value (Mafart model). In order to evaluate whether the selected z-value affects the parameter estimates, it was fixed to the minimum (10.4 °C for Bigelow, 11.37 °C for Mafart) and maximum (11.47 °C for Bigelow, 12.77 °C for Mafart) values estimated based on dynamic data. Moreover, because for the Mafart model the minimum z-value was estimated from isothermal experiments, it was also fixed to this value (10.87 °C).

Figure 3 and Figure 4 compare the estimated values of D_95_ and δ_95_, respectively, using different z-value. The goodness of the fit was evaluated using the parameters introduced in previous sections. Fixing the value of z resulted in a slight increase of the bias and accuracy factor (below 5% in every case) with respect to the procedure when it was estimated. Hence, fixing the z-value does not imply a significant loss of accuracy in the cases studied. However, this procedure results in a dramatic reduction of the estimated standard deviation of the D_95_ and δ_95_ (Figure 3 and Figure 4). Moreover, a relationship is observed between the value of z and the estimated D_95_ or δ_95_; using a higher value of z results in a lower estimated value of D_95_ and δ_95_. This is due to the correlation between both parameters for dynamic experiments, as has been reported in several studies [43,44]. Therefore, in this case, fixing the z-values reduces the uncertainty of the parameter estimates allowing a better evaluation of the impact of the different inactivation profiles on the heat resistance of *A. acidoterrestris*.

The results of this investigation show that, although most inactivation usually takes place during the holding phase, thermal inactivation processes should consider every stage. Even though the proportion of inactivation that takes place during the heating phase is relatively small, this phase can affect the thermal resistance of the microorganism during posterior phases. As shown in this investigation, the thermal resistance of *A. acidoterrestris* was reduced when the heating rate was between 20 and 30 °C/min. Therefore, the intensity of the treatment could potentially be reduced without affecting the safety of the product. Although the final recommendation is the same, the results of this investigation are different from those observed in similar experiments for *Escherichia coli* [18,45], *Salmonella* spp. [46], or *Listeria monocytogenes* [16,46] where the thermal resistance of the microorganism for high heating rates was the same as observed for isothermal experiments and slow heating rates induced a stress adaptation. This could be associated with the fact that those experiments were done using vegetative cells whereas this study analyzed *A. acidoterrestris* spores. It is possible that heating rates between 20 and 30 °C/min induce a physiological change in the spores, reducing their resistance compared to the ones of spores that are either heated immediately (isothermal treatment) or heated up at a slower rate. However, the heat resistance of bacterial spores under dynamic conditions has not yet been studied in enough detail, and additional experimental data are required to test this hypothesis. Nevertheless, the conclusions of this study can aid in improving thermal treatments of pasteurized juices and beverages to obtain safe products with a minimum impact on the sensorial quality and nutrient content of the products. Our results will contribute to establishing effective preservation treatments for fruit juices, taking into consideration the impact of the heating rates and heating profiles on *A. acidoterrestris*.

## 4. Conclusions

The results of this research evidence that, even though most of the inactivation occurs during the holding phase, thermal treatments should be designed to account for every step of the process. As concluded in this scientific work, heating rates between 20 and 30 °C/min during the heating phase reduced the resistance of *A. acidoterrestris* with respect to the one observed under isothermal conditions or with a heating rate of 1 °C/min. Therefore, more effective heat treatments can be attained through changes in the heating phase without modifying the temperature or duration during the holding phase. These results can significantly contribute to establishing effective preservation treatments to improve the microbial safety and stability of industrial thermal processes. Nevertheless, there is little information in the scientific literature regarding the inactivation of bacterial spores under dynamic heat treatments. Therefore, further studies on continuous heating systems and at higher heating rates are needed to determine the behavior and the effect of the heating rates under more realistic industrial situations.

## Figures and Tables

**Figure 1 foods-10-01496-f001:**
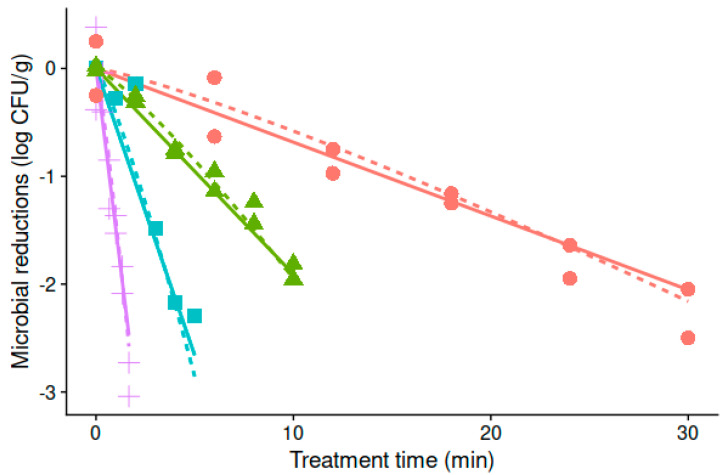
Survival curves of *A. acidoterrestris* spores on acidified peptone water at 90 (o), 95 (∆), 100 (□), and 105 °C (+). The lines show the fitted curves of the Bigelow (-) and Mafart (--) models following a one-step approach.

**Figure 2 foods-10-01496-f002:**
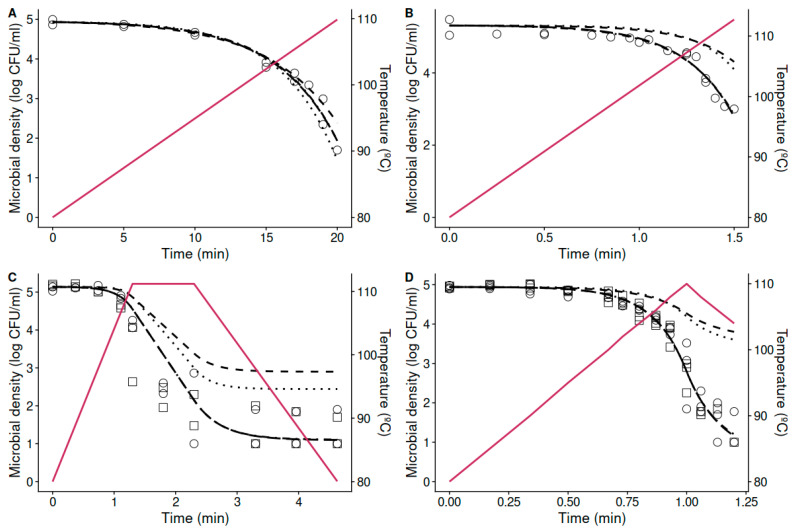
Survival curves of *Alicyclobacillus acidoterrestris* under heating rates at: 1 °C/min (**A**) and at 20 °C/min (**B**) in acidified peptone water (○); and under complex thermal treatments: complete treatment (**C**) and heating + cooling (**D**) in acidified peptone water (pH 3.5) (○) and orange juice (□). The lines show the temperature profile (solid line), the model predictions based on isothermal conditions for the Bigelow (- -) and Mafart (··) models, as well as the fits of the Bigelow (– –) and Mafart (-·-) using dynamic models.

**Figure 3 foods-10-01496-f003:**
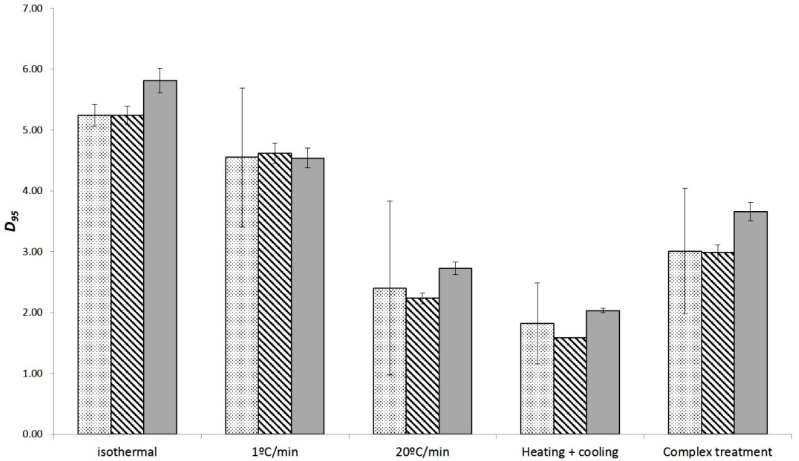
Estimated values of D_95_ for the different thermal treatments depending on the z-value used: adjusted to the non-isothermal results (dotted column), maximum estimated value (stripped column), or minimum estimated value (grey columns). The error bars represent the standard error of the parameter estimates.

**Figure 4 foods-10-01496-f004:**
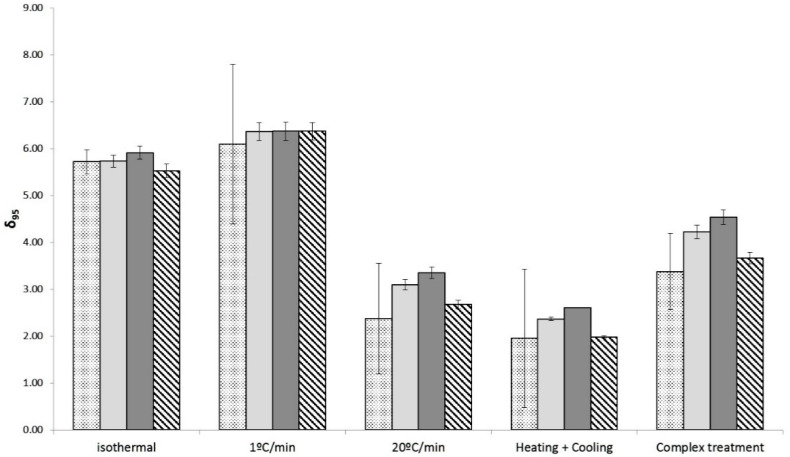
Estimated values of δ_95_ for the different thermal treatments depending on the z-value used: adjusted to the non-isothermal results (dotted column), maximum estimated value (stripped column), minimum estimated value (dark grey columns), or isothermal estimated value (light grey columns). The error bars represent the standard error of the parameter estimate.

**Table 1 foods-10-01496-t001:** Statistical indexes quantifying the precision of the predictions generated using the Bigelow and Mafart models for dynamic profiles based on model parameters estimated from isothermal experiments.

	*RMSE*		$A_{f}$		$B_{f}$		*AIC_c_*	
	Bigelow	Weibull	Bigelow	Weibull	Bigelow	Weibull	Bigelow	Weibull
1 °C/min	0.27	0.30	1.86	1.98	1.33	0.74	28.03	30.21
20 °C/min	0.71	0.67	5.18	4.68	3.72	3.59	44.66	45.64
Heating + Cooling	1.15	1.08	14.09	12.02	5.59	5.24	212.62	204.48
Complex treatment	1.28	1.05	19.17	11.12	10.39	6.39	129.72	111.89

**Table 2 foods-10-01496-t002:** Model parameters of the Bigelow model for inactivation *Alicyclobacillus acidoterrestris* spores in acid media obtained by model fitting of non-isothermal experiments. Results are reported as estimate ± SEM.

	D_95_ (min)	*z* (°C)	*RMSE*	$A_{f}$	$B_{f}$	*AIC_c_*
1 °C/min	5.77 ± 1.69	11.47 ± 2.03	0.20	1.58	1.04	27.61
20 °C/min	4.11 ± 2.00	10.79 ± 1.94	0.17	1.49	1.11	36.51
Heating + Cooling	2.39 ± 1.02	10.40 ± 1.80	0.25	1.78	1.03	129.59
Complete treatment	3.41 ± 1.16	11.18 ± 1.89	0.63	4.28	1.39	84.79

**Table 3 foods-10-01496-t003:** Model parameters of the Weibull model for inactivation *Alicyclobacillus acidoterrestris* spores in acid media obtained by model fitting of non-isothermal experiments. Results are reported as estimate ± SEM.

	$\delta_{95}\;(\min)$	*z* (°C)	*p*	*RMSE*	$A_{f}$	$B_{f}$	*AIC_c_*
1 °C/min	5.06 ± 0.66	10.87 ± 2.40	1.10 ± 0.07	0.20	1.58	1.01	29.62
20 °C/min	3.44 ± 1.00	12.22 ± 2.03	1.24 ± 0.14	0.18	1.50	1.12	38.52
Heating + Cooling	2.29 ± 0.90	12.77 ± 2.05	1.24 ± 0.14	0.25	1.78	1.02	131.64
Complete treatment	3.89 ± 1.01	12.45 ± 1.84	1.22 ± 0.14	0.63	4.28	1.39	86.82

## Data Availability

Data available on request due to restrictions of privacy.

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
