# Peer review of "Impact of Heating Rates on Alicyclobacillus acidoterrestris Heat Resistance under Non-Isothermal Treatments and Use of Mathematical Modelling to Optimize Orange Juice Processing"

_foods, 2021, doi:10.3390/foods10071496_

Round 1

Reviewer 1 Report

Manuscript ID: foods-1224596

Submitted for publication to foods

Title: Impact of heating rates on Alicyclobacillus acidoterrestris heat resistance under non-isothermal treatments and use of mathematical modelling to optimize food preservation

As the author stated, the objective of the current manuscript was to study “the effect of heating rates and complex thermal treatments on the inactivation kinetics of A. acidoterrestris”. I am not sure if the conclusions drawn can be solely attributed to the particular heating patterns employed. Do D95°C and z values reported, especially from non-isothermal data, refer exclusively to A. acidoterrestris spore inactivation? Is it possible that spores could be either directly inactivated, or the temperature (profile) applied could induce spore germination before their subsequent inactivation? In such case, a composite mathematical model, associated with a more demanding experimentation protocol, should be developed. In an earlier paper from most of the same authors, (Conesa, R., Andreu, S., Fernández, P. S., Esnoz, A. & Palop, A. Non-isothermal heat resistance determinations with the thermoresistometer Mastia. J. Appl. Microbiol. 2009, 107, 506-513), the authors concluded that “Under nonisothermal conditions, inactivation was reasonably well predicted from isothermal data”. What differentiates the current conclusions from the earlier findings?

How was the cooling of the samples achieved after the heat treatments that had not a cooling step?

Referring to the R package without describing the functions and the algorithm used leaves an important gap to all the conceptual approach.

As for the other statistical indices used, provide equation/procedure for the calculation of parameter ?, the likelihood of the model fit, appearing in (Eq. (9).

Does Eq. (6) come out of Eq. (4) or vice versa?

In Tables 2 and 3, state what numbers within the parentheses represent (which kind of uncertainty, SD, SE, 95% Confidence Intervals, or what?). Similarly, explain the error bars in the Figures.

On Figure 1, it will be better to show the regression lines instead of using lines to connect the experimental data. It will be also helpful to show the regression equations. Note that the D values (and their confidence intervals) for each isothermal treatment are not reported in the manuscript.

Comment on the enormous error depicted on Fig. (1), especially for the 100°C data (in some cases greater than ±1.5 log CFU/mL).

Are temperatures shown on Figure 2 the nominal temperatures set in the Mastia thermoresistometer? Were the actual temperatures achieved during the experiments continuously monitored and recorded? Were there any “delays” in reaching the set temperatures? Could this affect calculations?

What was the spore detection limit? Figure 1 data should have been taken for longer times and spore reduction to lower than 2 or 3 log CFU/mL should have been taken.

In Figure 2, you show the inactivation curves obtained using parameters from isothermal data. Although in line 274, you refer to this Figure, I think that another, very important figure is missing, where you should show the same inactivation curves (for all profiles used), but based on kinetic parameters calculated on the basis of non-isothermal data. Since this is a major statement that you use for your conclusions, I think that is crucial to provide such a figure and critically comment on it, by comparing it with Figure 2.

Why did you decide to repeat the calculations for different z-values, fixing the other parameters to known values? This point needs justification. A particular non-linear temperature profile might influence the z value to a greater extend that it influences the DTref value. On that note, the selected reference temperature could also affect the results.

Please, explain in more details your statement in line 337/338 (“This indicates structural identifiability issues of the model”).

Syntax/grammatical errors/formatting:

       Re-consider carefully the text, in terms of editing

       Keep Tables and their captions on same page

       Check Equation number format

       Equation (8) missing (need for re-numbering)

       Figure1format

       Alicyclobacillus acidoterrestris must be written in italics, throughout the text

Author Response

The authors are grateful to the reviewer for his/her comments and valuable contributions that have helped to improve the first version of the manuscript. We provide a list of replies and a description of the changes made to the new version.

Title: Impact of heating rates on Alicyclobacillus acidoterrestris heat resistance under non-isothermal treatments and use of mathematical modelling to optimize food preservation

As the author stated, the objective of the current manuscript was to study “the effect of heating rates and complex thermal treatments on the inactivation kinetics of A. acidoterrestris”. I am not sure if the conclusions drawn can be solely attributed to the particular heating patterns employed. Do D95°C and z values reported, especially from non-isothermal data, refer exclusively to A. acidoterrestris spore inactivation? Is it possible that spores could be either directly inactivated, or the temperature (profile) applied could induce spore germination before their subsequent inactivation? In such case, a composite mathematical model, associated with a more demanding experimentation protocol, should be developed.

     We acknowledge the interesting comment by the reviewer. Considering the temperature profile, it is very unlikely that the spores would go through germination considering the slowest time/temperature profile used (1ºC/min, from 80 to 100ºC). In the rest, the heating rates were much faster and reached higher temperatures. F. eg. Collado et al (2006) found that heat shocked B. cereus spores had a higher germination rate than non-heat-shocked ones, and times  lower than 30 min did not induce germination, even in the presence of L-alanine as germinant. Kohler et al (2018) also identified activation at temperatures in the range of 65-80ºC for relatively short times but no germination. For A. acidoterrestris spores, germination studies used temperatures of 80ºC for 10-20 min to activate spore suspensions, but they did not germinate until appropriate recovery conditions were met (f.eg. specific germinants or broth media and incubation temperatures of 45-46ºC) (Terano et al., 2005, Xu et al., 2019). We have mentioned these aspects and included references in the new version to clarify it.

References mentioned (some of them are in the new manuscript):

  • Collado et al. 2006. Modelling the effect of a heat shock and germinant concentration on spore germination of a wild strain of Bacillus cereus. J. Food Microbiol. 106, 85-89. DOI: 10.1016/j.ijfoodmicro.2005.06.009
  • Kohler et al. Incorporating germination-induction into decontamination strategies for bacterial spores. J. Appl. Microbiol., 124(1):2-14. doi: 10.1111/jam.13600
  • Terano H., Takahashi K., Sakakibara Y. 2005. Characterization of spore germination of a thermoacidophilic spore-forming bacterium, Alicyclobacilus acidoterrestris. Biotechnol. Biochem. 69, 1217-1220.
  • Xu X., Ran J., Jiao L., Liang X., Zhao R. 2019. Label free quantitative analysis of Alicyclobacillus acidoterrestris spore germination subjected to low ambient pH. Food Res. Int. 115, 580-588.

In an earlier paper from most of the same authors, (Conesa, R., Andreu, S., Fernández, P. S., Esnoz, A. & Palop, A. Non-isothermal heat resistance determinations with the thermoresistometer Mastia. J. Appl. Microbiol. 2009, 107, 506-513), the authors concluded that “Under nonisothermal conditions, inactivation was reasonably well predicted from isothermal data”. What differentiates the current conclusions from the earlier findings?

     We appreciate the comment by the reviewer. The experiments reported in Conesa et al. (2009) used heating rates of 1 and 20ºC/min. They can be considered as preliminary, since they were descriptive results showing the ability of the equipment to perform non isothermal studies. In this research, at heating rates of 1ºC/min we also observed a good agreement between the isothermal prediction and dynamic data; whereas bias was observed for the higher heating rate (20ºC/min). In the previous manuscript we used the term “inactivation was reasonably well predicted” as there was no other information to compare. Therefore, our results are consistent with those by Conesa et al. as a starting point, and they provide a complete study of all the possible combinations of non-isothermal treatments that can be applied, simulating industrial conditions. Additionally, an advanced modelling approach is used. This information has been included in the revised version of the manuscript, as it was not properly discussed in the first version.

How was the cooling of the samples achieved after the heat treatments that had not a cooling step?

For all the samples, they were immediately cooled in an ice-water bath after collection in the test tubes. This information has been included in the methodology (L 146).

Referring to the R package without describing the functions and the algorithm used leaves an important gap to all the conceptual approach.

We acknowledge the comment and we have added the relevant algorithms in the revised version of the manuscript.

As for the other statistical indices used, provide equation/procedure for the calculation of parameter ?, the likelihood of the model fit, appearing in (Eq. (9).

In the revised version, the likelihood has been substituted by its expression based on logN.

Does Eq. (6) come out of Eq. (4) or vice versa?

This part has been modified in the revised version, following a more logical order (presenting the isothermal model first).

In Tables 2 and 3, state what numbers within the parentheses represent (which kind of uncertainty, SD, SE, 95% Confidence Intervals, or what?). Similarly, explain the error bars in the Figures.

The notation in the tables has been changed to the more usual “estimate + SEM”. This change has been reflected in the table captions.

On Figure 1, it will be better to show the regression lines instead of using lines to connect the experimental data. It will be also helpful to show the regression equations. Note that the D values (and their confidence intervals) for each isothermal treatment are not reported in the manuscript.

We acknowledge the reviewer’s comment and the plot has been updated accordingly including the models fitted. Moreover, the error bars have been substituted by the actual observations, as they show better the spread of the data.

Note however that the isothermal data have been fitted using a one-step approach. This approach does not fit a single survivor curve for each temperature, providing instead a global fit to the whole data based on a D-value (or delta) and a z-value. This point has been clarified in the revised version of the manuscript.

Comment on the enormous error depicted on Fig. (1), especially for the 100°C data (in some cases greater than ±1.5 log CFU/mL).

     We acknowledge the reviewers comment, as there was an error when making this plot. The error bars have been substituted by the actual observations in the revised version of the manuscript, showing that there is a good agreement between model predictions and observations.

Are temperatures shown on Figure 2 the nominal temperatures set in the Mastia thermoresistometer? Were the actual temperatures achieved during the experiments continuously monitored and recorded? Were there any “delays” in reaching the set temperatures? Could this affect calculations?

     The Mastia thermoresistometer provides a record of the temperature in the media during the experiment. These records were compared against the programmed ones for each repetition of the experiment, observing insignificant differences. As an additional check, model predictions under dynamic conditions were calculated using both the measured and programmed profiles, without observing any relevant difference. We decided to include in the plots the programmed profiles because we believe they represent better the type of treatment.

This information has been clarified in the revised version of the manuscript.

What was the spore detection limit? Figure 1 data should have been taken for longer times and spore reduction to lower than 2 or 3 log CFU/mL should have been taken.

     The spore detection limit was 1 log10 CFU/ml. Every isothermal experiment performed resulted in, at least, 2 log reductions (and 3 in most cases) of the microbial concentration, a value that is usually considered acceptable in predictive microbiology. It is true that longer treatment times may deviate from the linear trend observed in our experiment (e.g. due to tailing). However, our dynamic treatments are in the same range of reductions, so this effect should not be relevant under dynamic conditions either.

In Figure 2, you show the inactivation curves obtained using parameters from isothermal data. Although in line 274, you refer to this Figure, I think that another, very important figure is missing, where you should show the same inactivation curves (for all profiles used), but based on kinetic parameters calculated on the basis of non-isothermal data. Since this is a major statement that you use for your conclusions, I think that is crucial to provide such a figure and critically comment on it, by comparing it with Figure 2.

We acknowledge the reviewer’s comment, as this information was missing in the original version of the manuscript. The models fitted to dynamic condition have been included in Figure 2. We have decided not to include a new figure because the article already has considerable graphical content (figures and tables). The text in the R&D section has been updated accordingly.

Why did you decide to repeat the calculations for different z-values, fixing the other parameters to known values? This point needs justification. A particular non-linear temperature profile might influence the z value to a greater extend that it influences the DTref value. On that note, the selected reference temperature could also affect the results.

We decided to fix the z-value because, unlike for the D(delta)-value, there were no significant differences between the values of this parameter estimated under dynamic conditions (Tables 2 & 3) or isothermal conditions. We selected the maximum and minimum value to provide an estimate of how this decision affects the parameter estimates, as there is a correlation between the z-value and the D-value. We agree that this approach may not be generalized to other conditions (microorganisms, media) because, as mentioned above, the effect of dynamic profiles on heat inactivation depends on a variety of factors. Nonetheless, we believe that in the case of this microorganism and these media this approach is justified and provides the best information. 

This hypothesis has been discussed further in the revised version of the manuscript.

Regarding the reference temperature, it is unlikely that the reference temperature affects the parameter estimates in this case. As shown in Tables 2 and 3, the z-value remains constant between the different temperature profiles, so the secondary models are “shifted” in the y-direction. Therefore, a change in the reference temperature should not affect the differences between the values of logD estimated under different conditions. Nonetheless, this was tested during the analysis, without observing any different in the results.

Please, explain in more details your statement in line 337/338 (“This indicates structural identifiability issues of the model”).

This sentence has been rewritten, providing further information and references. It reads (L 437 onwards) “This is due to the correlation between both parameters for dynamic experiments, as has been reported in several studies”.

Syntax/grammatical errors/formatting:

       Re-consider carefully the text, in terms of editing

We acknowledge the comment by the reviewer and we apologize for these editing mistakes. In the revised vesion, the text has been thoroughly revised improving the writing. Some of the mistakes were due to changes that appeared when using the template, we hope that everything will appear correctly this time.

       - Keep Tables and their captions on same page

Tables and captions have been kept in the same page.

     - Check Equation number format       Equation (8) missing (need for re-numbering)

Equations and their numbering have been revised.

      - Figure 1 format

Figure 1 has been modified according to the comments of the reviewer and the format has been improved.

      - Alicyclobacillus acidoterrestris must be written in italics, throughout the text.

It has been changed in the text accordingly.

Reviewer 2 Report

The purpose of this article was to employ a modelling approach using different heating rates in order to elucidate the inactivation dynamics of Alicyclobacillus acidoterrestrisunder non-isothermal treatments. Overall, the work is interesting and the experimental design takes into account several heat treatment scenarios for the processing of the heating media (acidified peptone water and orange juice). My only concern for this work is about the performance of the models that present a tendency to overestimate the population of the bacterium, especially when the heating rate of the temperature profile is increased. This is well exemplified by the survival curves in Figure 2 and also by the very high (in some cases) values of the bias and accuracy factors. My comments are below:

Title: the wording “… to optimize food preservation” does not reflect the content of this work as the target food was orange juice and not “food” in general and also there was no preservation considered in this work but rather processing. I think it would be more accurate to write “…to optimize orange juice processing”.

Page 1, Line 22: please correct “estability” to “stability”

Page 2, Line 64: please use “growth” instead of “outgrowth”

Page 3, Line 110: Please indicate the Brix of the orange juice. Was it pasteurized? Did you check for the presence of the bacterium before the beginning of experiments?

Page 5, Equations 10 and 11: These equations for the bias and accuracy factors are not correct. Please consult Ross (1996), Indices for performance evaluation of predictive models in food microbiology, J. Appl. Bacteriol., 81, 501-508.  Otherwise provide a reference where these equations have been taken from.

Page 7, lines 235-237: Is there any biological explanation why the bias of the model predictions increases when the heating rate of the temperature profile also increased? For example, from Figure 2D it seems that the greatest deviations are observed during the cooling phase and also in Figure 2C during the holding and cooling phase. Could the authors provide some explanation as far the biology of the bacterium is concerned?

Author Response

The authors are grateful to the reviewer for his/her comments and valuable contributions that have helped to improve the first version of the manuscript. We provide a list of replies and a description of the changes made to the new version.

The purpose of this article was to employ a modelling approach using different heating rates in order to elucidate the inactivation dynamics of Alicyclobacillus acidoterrestris under non-isothermal treatments. Overall, the work is interesting and the experimental design takes into account several heat treatment scenarios for the processing of the heating media (acidified peptone water and orange juice). My only concern for this work is about the performance of the models that present a tendency to overestimate the population of the bacterium, especially when the heating rate of the temperature profile is increased. This is well exemplified by the survival curves in Figure 2 and also by the very high (in some cases) values of the bias and accuracy factors. My comments are below:

We acknowledge the constructive comments by the reviewer. Nonetheless, we would like to emphasize that the overestimation illustrated in Figure 2 is based on the predictions of the model fitted to data gathered under isothermal conditions. Indeed, these figures serve to prove that the heating rate has an impact on the microbial response to the thermal treatment.

For that reason, we also fit the models to the dynamic data directly, obtaining a sufficiently good fit (Bf~1; Table 2-3). Then, the conclusions drawn in section 3.3 are based on these models, so they are not affected by the bias mentioned by the reviewer.

This misunderstanding is our fault because the original version of the manuscript did not include the models fitted under dynamic conditions in this plot. In the revised version of the manuscript, Figure 2 has been modified, including also the models fitted to the dynamic data.

Title: the wording “… to optimize food preservation” does not reflect the content of this work as the target food was orange juice and not “food” in general and also there was no preservation considered in this work but rather processing. I think it would be more accurate to write “…to optimize orange juice processing”.

The title has been changed according to the comment of the reviewer.

Page 1, Line 22: please correct “estability” to “stability”

We have modified the sentence in the new version and the word does not appear in the abstract.

Page 2, Line 64: please use “growth” instead of “outgrowth”

It has been modified as indicated.

Page 3, Line 110: Please indicate the Brix of the orange juice. Was it pasteurized? Did you check for the presence of the bacterium before the beginning of experiments?
     Pasteurized commercial orange juice (García Carrión, Jumilla, Spain) was used. It had 11.0 °Brix. Sterility checks were performed before the beginning of the experiments, denoting no presence of viable microorganisms. This information has been included in the text.

Page 5, Equations 10 and 11: These equations for the bias and accuracy factors are not correct. Please consult Ross (1996), Indices for performance evaluation of predictive models in food microbiology, J. Appl. Bacteriol., 81, 501-508.  Otherwise provide a reference where these equations have been taken from.

We acknowledge the reviewer’s comment. They have been updated (they were implemented correctly in the code). They are now Eq. 9 and 10.

Page 7, lines 235-237: Is there any biological explanation why the bias of the model predictions increases when the heating rate of the temperature profile also increased? For example, from Figure 2D it seems that the greatest deviations are observed during the cooling phase and also in Figure 2C during the holding and cooling phase. Could the authors provide some explanation as far the biology of the bacterium is concerned?

We acknowledge the interesting comment from the reviewer. The discussion has been revised in the updated version of the manuscript providing an explanation for these effects in the last paragraph of the Results and Discussion section. It now appears as: “It is possible that heating rates between 20 and 30ºC/min induce a physiological change in the spores reducing its resistance with respect to the ones of spores that are either heated immediately (isothermal treatment) or are heated up with a slower rate. However, the heat resistance of bacterial spores under dynamic conditions has not yet been studied in enough detail, and additional experimental data is required to test this hypothesis. Nevertheless, the conclusions of this study can aid towards improving thermal treatments of pasteurized juices and beverages to obtain safe products with a minimum impact on the sensorial quality and nutrient content of the products. Our results will contribute to establish effective preservation treatments for fruit juices, taking into consideration the impact of the heating rates and heating profiles on A. acidoterrestris.”

Round 2

Reviewer 1 Report

I believe that the authors adequately addressed reviewers’ comments. The following points should be though further considered before publication (line numbers refer to the CLEAN revised manuscript):

Line 140: The following statement must be replaced by Eq. number: Error! Reference source not found.

Equation number in Eq. (5) is out of place.

Equation (8) starts with a parenthesis and its format must be corrected.

Line 234: are biased instead of is biased.

Figure 2 caption must be reconsidered. Furthermore, I recommend making the data markers bigger, the lines thicker, and letter font larger.

Lines 332-335 should be omitted.

In reference #26, “J.l Infect. Dis.”, seems wrong.

Author Response

We would like to acknowledge de revision and comments made by the reviewer, that have helped to improve significantly the clarity and focus of the paper. We reply to the individual comments below.

I believe that the authors adequately addressed reviewers’ comments. The following points should be though further considered before publication (line numbers refer to the CLEAN revised manuscript):

Line 140: The following statement must be replaced by Eq. number: Error! Reference source not found.

We have changed the Error statement to Equation (3), it seems to be a compatibility problem as we do not always see it.

Equation number in Eq. (5) is out of place.

We have replaced the number in Eq. (5) accordingly.

Equation (8) starts with a parenthesis and its format must be corrected.

We have deleted the initial parenthesis and corrected the format as indicated.

Line 234: are biased instead of is biased.

It has been changed as proposed.

Figure 2 caption must be reconsidered. Furthermore, I recommend making the data markers bigger, the lines thicker, and letter font larger.

Figure caption has been changed, it now reads:

Figure 2. Survival curves of Alicyclobacillus acidoterrestris under heating rates at: 1 ºC/min (A) and at 20 ºC/min (B) in acidified peptone water (â—‹). And under complex thermal treatments: complete treatment (C) and heating + cooling (D), in acidified peptone water (pH 3.5) (â—‹) and orange juice (â–¡).  The lines show the temperature profile (solid line), the model predictions based on isothermal conditions for the Bigelow (- -) and Mafart (··) models, as well as the fits of the Bigelow (– –) and Mafart (-·-) using dynamic models.

Figure 2 has also been updated following the recommendations of the reviewer.

Lines 332-335 should be omitted.

The paragraph indicated has been deleted as indicated.

In reference #26, “J.l Infect. Dis.”, seems wrong.

The letter “I” has been deleted to leave the reference correctly written.